# Tourism Destination Marketing: Academic Knowledge

**Marios Sotiriadis**

Ningbo University-University of Angers Joint Institute/Sino-European Institute of Tourism and Culture, Ningbo University, Ningbo 315211, China; sotiriadis@mbu.edu.cn or sotermarios@outlook.com

**Definition:** A holistic, multi-organization view of marketing or destination management organizations (DMOs) who must muster the best efforts of many partner organizations and individuals (stakeholders) to have the greatest success. Destination marketing is described as "a continuous, sequential process through which a DMO plans, researches, implements, controls and evaluates programs aimed at satisfying tourists' needs and wants as well as the destination's and DMO's visions, goals and objectives". The effectiveness of marketing activities depends on the efforts and plans of tourism suppliers and other entities. This definition posits that marketing is a managerial function/domain that should be performed in a systematic manner adopting and implementing the appropriate approaches, as well as suitable tools and methods. In doing so, it is believed that a tourism destination (through the organizational structure of a DMO) can attain the expected outputs beneficial to all stakeholders, i.e., the tourism industry, hosting communities/populations, and tourists/visitors. The effective implementation of tourism destination marketing principles and methods constitutes an efficient and smart pillar, a cornerstone to attain a balance/equilibrium between the perceptions and interests, sometimes conflicting, of stakeholders by minimizing the negative impacts and maximizing the benefits resulting from tourism. All the same, it is worth noting that marketing is not a panacea, nor a kind of magic stick.

**Keywords:** marketing; tourism destination; managerial cycle; marketing planning; strategies; marketing action plans





## 1. Introduction

Over the last three decades, the business environment and markets in tourism and travel industries have evolved and been significantly changed due to a series of factors, such as globalization, volatile markets, highly intense competition, crises of all kinds, and widespread diffusion of information and communication technologies [1]. Academic literature indicates that all these factors—shifters and drifters—are considerably influencing tourists' consumption behavior, as well as business functions and processes of tourism suppliers and destinations [2,3].

Against this background, the volume of published research focusing primarily on issues of tourism destination management and marketing has been steadily growing over the years. The focus of this entry paper is on the research area of destination marketing. The article's purpose is twofold: (i) to perform a synthesis of the wealth of academic research published over the last three decades and (ii) to suggest pathways/avenues for future research. This synthesis allows us to identify and highlight the key elements of academic knowledge in this field. The following points provide a definition of the topic and briefly present the approach to and methodology implemented in performing the synthesis of existing literature.

By outlining and synthetizing what is known in this research field, this entry paper lays the groundwork, providing a timely insight into the current state of research on marketing of tourism destinations. Through a systematic quantitative literature review of articles published in tourism-related journals, this is achieved through meeting the article's

aim. This outline of existing knowledge provides opportunities, directions, and avenues for future research in this increasingly important domain/area of tourism.

The approach to this entry is as follows. Marketing is a very broad research field; the same stands for the area of tourism destination marketing [1,4]. It is believed that to better consider and synthesize the existing knowledge in this area, we have to adopt a systematic approach aimed at structuring and classifying the wealth and variety of academic knowledge. This article argues that the most adequate approach is to consider the topic in terms of marketing management, allowing us to achieve a better perception and acquire a more integrative image.

According to Kotler and Keller [5], "Marketing management is the analysis, planning, implementation and control of programs designed to bring about desired exchanges with target audiences for the purpose of personal and of mutual gain. It relies heavily on the adoption and coordination of product, price, promotion and place for achieving responses." Hence, marketing management is a business process of managing marketing activities. Marketing management decisions are based on strong knowledge of marketing functions and clear understanding and application of managerial methods and techniques for decision-making [3,5,6]. Therefore, this article considers and synthesizes the marketing knowledge/existing literature as a managerial process.

Methodology: This entry adopted and implemented the following methodology. Given that the aim of this article was to outline the current state of knowledge on tourism destination marketing, the best suited method to address this aim is the systematic quantitative review. The type of review is systematic as the methods used to survey and select the papers are explicit and reproducible [7]. The four-step systematic quantitative literature review process consists of (i) determining review aim; (ii) identifying search terms, databases, and literature selection criteria; (iii) searching the databases for the literature and screening search outcomes against the criteria before refining exclusion and inclusion criteria; and (iv) appraising literature quality and relevance, structuring summary tables through extracting relevant information. Given this study's review aim, the search strings "tourism destination" OR "marketing destination" OR "tourism promotion" were used in titles, keywords, and abstracts to search for relevant literature firstly in the Scopus academic database, followed by four additional databases: EBSCO, Elsevier, ProQuest, and Emerald. To safeguard the quality and effectiveness of the review, only original research articles published in English-language peer-reviewed journals were considered. The search was time-bound for the last three decades, since 1990.

## 2. Current State of Academic Research in Destination Marketing

### 2.1. Topics and Issues

Academic literature suggests that the research field of marketing of tourism destinations has made leaps over the last three decades and achieved a stage of maturity [6,8,9]. As a result, a solid and robust knowledge has been built up. This article elaborates on a classification of research areas/specific domains based on the main components and activities of the role of marketing and promotion of a destination management organization (DMO) as well as the two ends of the supply/value chain of tourism destination [1]. The following classification of the research topics of tourism destination marketing into six headings/domains is therefore suggested (Table 1).

Overall, this review covers 227 journal articles published over three decades, from 1990 to 2020. Apparently, the range of topics and issues are considerable. It can be easily understood why the research field of tourism destination marketing has been established and constitutes a knowledge body on its own. This development is documented by a number of academic journals publishing articles related to tourism marketing.

**Table 1.** A classification/synthesis of topics in the research field of tourism destination marketing.

| Destination Marketing Functions: Main Components and Activities | Topics and Issues | Focus On | Volume of Studies and Examples |
|---|---|---|---|
| 1. Marketing planning and market research | Marketing research<br>Marketing environmental analysis<br>Marketing information system<br>Destination marketing system | Effective and efficient planning<br>Strategic analyses, techniques and methods<br>Process, influencing factors and results. | 30 studies [10–39]<br>Examples:<br>Frew and O'Connor [19]<br>Murdy and Pike [29]<br>Hay and Yeoman [22]<br>Marais et al. [25] |
| 2. Marketing strategies: design/elaboration of main marketing strategies | Market segmentation analysis<br>Targeting and Positioning<br>Destination image<br>Destination branding | Effectiveness and appropriateness of marketing choices<br>Positioning-image-branding approach<br>Brand development (process) and management<br>Loyalty | 69 studies [40–109]<br>Examples:<br>Dolnicar and Grün [55]<br>Orth and Tureckova [82]<br>Kendall and Gursoy [67]<br>Garcia et al. [62] |
| 3. Implementation of strategies: marketing action plans | Marketing mix<br>Usage of tools to communicate with targeted markets<br>Digital marketing<br>Social media marketing | Content, media, and techniques.<br>The 8Ps model (product, price, place/distribution, promotion, packaging, programming, partnership, and people).<br>Assessing the appropriateness (suitable use) and effectiveness of various tools and media. | 44 studies [110–153]<br>Examples:<br>Alberca et al. [110]<br>Byun and Jang [114]<br>Dore and Crouch [116]<br>Hays et al. [124]<br>Jiang et al. [127] |
| 4. Control and evaluation: monitoring and assessing performance | Customer feedback<br>Performance evaluation<br>Benchmarking<br>Measurement of the effectiveness of marketing activities | Control methods and metrics<br>Monitoring and assessing progress<br>Evaluation procedures and measurements<br>Performance measures/metrics for the action plans and activities included in the marketing plan | 17 studies [154–170]<br>Examples:<br>Faulkner [156]<br>Kulendran and Dwyer [159]<br>Li and Wang [161]<br>Zavattaro et al. [170] |
| Two ends of the value/supply chain | | | |
| 5. Tourist consumer behavior | Decision-making<br>Customer experience<br>Co-creation<br>External and internal influences<br>Influence of communications mix | Influence of the various media and tools<br>Effectiveness and efficiency of marketing activities | 32 studies [171–202]<br>Examples:<br>Chen and Gursoy [176]<br>Chiou et al. [178]<br>Jimenez-Barreto et al. [182]<br>Lam et al. [187]<br>Loda et al. [189] |
| 6. Organizational structure (DMO) | Organization, functions, and roles<br>Governance<br>Benchmarking | Marketing performance and effectiveness<br>Overall performance and return on investment<br>Efficiency and effectiveness<br>Accountability and reporting | 22 studies [203–224]<br>Examples:<br>Beritelli et al. [203]<br>Cox and Wray [207]<br>De Carlo [209]<br>Pike [218]<br>Ruhanen et al. [221] |
| – | Collaboration | Approaches and strategies | 13 studies [225–237]<br>Wang et al. [235]<br>Wang and Xiang [237] |

High-ranking academic journals are publishing articles fully related to this topic, such as *Journal of Destination Marketing and Management* or others closely related, such as

*Journal of Travel & Tourism Marketing*. Likewise, research articles related to destination marketing are published in other leading academic journals included in the Social Sciences Citation Index® (SSCI) List (under the subject category "Hospitality, leisure, sport & tourism"), namely: *Annals of Tourism Research*, *Asia Pacific Journal of Tourism Research*, *Current Issues in Tourism*, *International Journal of Tourism Research*, *Journal of Hospitality and Tourism Research*, *Journal of Hospitality and Tourism Management*, *Journal of Travel Research*, *Journal of Vacation Marketing*, *Tourism Analysis*, *Tourism Management*, *Tourism Management Perspectives*, *Tourism Review*, and *Tourism Studies*. These leading international journals seek to develop a robust theoretical understanding of all aspects of destination marketing, among other key topics of tourism research.

The variety and the scope of the published research is considerable and covers all above issues and aspects resulting in the creation of tourism marketing and tourism destination marketing knowledge [6]. Academic research has examined and explored extensively the topics and issues depicted in Table 1. A special mention should be given to some "hot topics", as follows.

Destination image and branding: Destination branding is an important research area and a powerful instrument for building the positioning of tourism destinations [96,238]. Branding tourism destinations is very challenging given the diversity of the destination mix and the experience characteristic of tourism [1]. The issues explored by scholars in this research area include place attachment, destination loyalty, and impact of Web 2.0 platforms [96].

Destination marketing planning includes the process, influencing factors, stakeholders, prerequisites and critical success factors still attracting the interest of academic research. It is recommended that the planning should be based on the principles and recommendations of strategic planning and management [7].

Integrated marketing communications and promotional mix: The effectiveness of communication actions is another topic due to high expenditure and investment.

Destination stakeholder collaboration: Tourist services are packages of tangible and intangible elements offered by various suppliers at destination level [36,239]. A major part of tourism offering components, such as cultural monuments and natural resources, are beyond managerial organization and control. At destination level, there is a need for coordinated action of the tourism industry and local stakeholders to design and promote an offering of experience opportunities.

Tourism is a fertile field for collaborations of all types. This is mainly because no single company, government agency, or other type of organization is able to control all the stages in the tourism value chain, while visitors expect destination offers to be integrated [226–228]. Therefore, one of the key roles of a DMO in the field of marketing is leadership and coordination, which involves setting the agenda for tourism and coordinating all stakeholders' efforts toward achieving that agenda. Effective destination marketing requires effort by other stakeholders within the destination. Collaboration with other organizations and individuals is a must [235–237]. DMOs can achieve much more for their destinations when they work in cooperation with others. Hence, a DMO should assume the role of the leader of tourism in its geographic location and coordinate the efforts of all the tourism stakeholders. This involves fostering cooperation among government agencies and within the private sector and building collaboration such as networks, alliances to reach specific goals. Public–private partnerships (PPPs) are one collaborative strategy that is becoming more popular in destination marketing.

Therefore, collaboration, various forms and strategies, and their interrelationship with other areas and functions of destination marketing, such as marketing activities, theming, public–private partnerships, alliances, and branding are imperative for promoting tourism experience opportunities at destination level [235–237,239].

## 2.2. Interrelationships with Other Research Theories, Disciplines, and Domains

### 2.2.1. Destination Management

Destination management and destination marketing are two highly interrelated research domains in tourism. In fact, destination marketing is one of the functions within the broader framework of destination management. The role of DMOs is to coordinate the management of tourism assets and resources. The main aim of marketing activities performed by DMOs is to increase and sustain tourism flows in the destination [1,240]. Therefore, the effectiveness of destination management is significantly influenced by the DMO's marketing activities. A review article by Ávila-Robinson and Wakabayashi [8]—covering a time period of 11 years, from 2005 to 2016—provides an integrated study of the structure and directions of destination management and marketing research. Their study identifies the main topics that include sustainable development, competitiveness, development and innovation, and ICTs, and significant interactions between management- and marketing-oriented research domains [8].

### 2.2.2. Consumer Behavior

The analysis of tourist behavior constitutes a fundamental topic to be considered by tourism destinations when planning marketing strategies. This stream of research encompasses issues such as the factors that affect tourists' preferences and choices, tourists' motivations, different types of tourists with a wide range of expectations and needs, understanding of tourists' travel decisions, and tourist market segmentation. Consumer behavior is one of the most researched areas in the field of tourism [241]. A review article examined the key concepts, external influences, and opportune research contexts. It is suggested that the key concepts in this area include decision-making, values, motivations, self-concept and personality, expectations, attitudes, perceptions, satisfaction, trust, and loyalty. There are three important external influences on tourism behavior, namely technology, Generation Y, and the rise in concern over ethical consumption.

Another interesting issue is the tourist experience as the essential element around which a competitive destination should be planned and developed. Related aspects include the components of tourists' experiences rendering them memorable and unforgettable. Finally, tourist behavior has been transformed with the advent and extensive use of communication and information tools such as the Internet, mobile devices, social networks, online booking engines having a considerable impact on decision-making [242].

### 2.2.3. ICTs and Smart Tourism

The growing importance of digital marketing for DMOs is undeniable. DMOs are now making very significant use of ICTs in their marketing activities. Digital marketing or interactive marketing involves working with all forms of ICTs that use digital formats and platforms. The six main components of DMO digital marketing are content creation, websites, search engine marketing and optimization, social media, apps, and e-mail marketing [1,240]. For DMOs, the Internet along with social media have become the major information dissemination and digital marketing tools. Academic research has focused on the effectiveness and influence of digital marketing channels and techniques. The main topic is the destination websites; their evaluations and designs are of major research interest [243].

The most recent developments resulted in smart tourism, which refers to the burgeoning phenomenon of the application of ICTs in developing innovative tools and approaches to improve tourism [244,245]. Inevitably, the growth of the smart tourism phenomenon has attracted attention from scholars attempting to explore related elements and issues [245,246]. The study by Mehraliyev et al. [247] points out that marketing topics are a significant component of current smart tourism research. Some scholars indicate that the service-dominant logic constitutes a valuable theoretical foundation for understanding the implications of smart tourism.

### 2.2.4. Service-Dominant Logic

The service-dominant logic (SDL) postulated by Vargo and Lusch [248,249] is a tourism-friendly evolution of marketing thought. The tourism industry gradually embraced marketing theories and principles of SDL. The main propositions of SDL and the links between this and related marketing concepts have been explored in the tourism field; see, for instance, McCabe [250]. Li [251] suggested that SDL provides a useful theoretical perspective in making sense of marketing practices in the field of tourism destination. In addition, the SDL research stance is considered as valuable approach to explain the value co-creation in different tourism contexts [252].

A recently emerged customer-dominant logic in marketing constitutes a significant theoretical contribution in the sense that it provides insights into tourists' co-creation practices. The study by Rihova et al. [253] aimed to explore specific customer-to-customer (C2C) co-creation practices and related value outcomes in tourism. The study highlights the importance of value formed when tourists cocreate with each other in tourism settings and suggests specific opportunities for facilitating this process.

### 2.2.5. Sustainability and Sustainable Tourism

Tourism marketing has typically been seen as exploitative and fueling hedonistic consumerism. Scholars suggest that the sustainability paradigm and principles can make a significant contribution in the field of marketing. Font and McCabe [254] point out that sustainability marketing can use marketing principles and methods effectively by understanding market needs, designing more sustainable offering and experience opportunities, and identifying more persuasive methods of communication to positively influence tourists' behavior [255,256]. It is therefore suggested that marketing could have a valuable contribution to making tourist destinations more attractive to visitors. Destination marketing has started paying attention to changing value systems regarding environmentally conscious tourism and sustainable development strategies [257]. The review article by Font and McCabe [254] explores sustainability marketing's two fundamental approaches, that of market development, using market segmentation, and that of sustainable product development. In this vein, a recent study by Shen et al. [258] explores the influence of social networking sites on adopting a sustainable and responsible behavior by tourists. It was found that the most significant influence is at pre-trip and during/on-site stages of the travel cycle.

### 3. Future Research Agenda: Suggested Pathways

As already indicated, academic research in this research field, driven by academic curiosity and industry issues, has made significant progress and advanced a body of knowledge, leading to an established field [6]. The synthesis of existing knowledge allows scholars understand the recent progress in this research field. Nevertheless, there are still issues and topics that require further investigation by conceptual and empirical endeavors. This section attempts to identify and outline suggested future research endeavors in the area of destination marketing.

It is estimated that the hedonic and affective aspects of consumer behavior research in tourism must be brought to bear on the wider consumer behavior and marketing literature [241]. Some issues such as trip satisfaction, experience, and destination loyalty require further investigation. Another largely unexplored area is the role of emotions and their treatment within experiential tourism offerings. The advances in consumer behavior research are expected to expand the interdisciplinarity of destination branding and to enrich its study with new concepts as well as the traditional constructs of loyalty, trust, involvement, and motivation [238]. Likewise, there is a requirement for in-depth analysis that addresses the challenges in destination branding [96].

Destination marketing has just started to exploit the potential of and analyze the risks of technological advances in the field of smart and enabling technologies, such as virtual and augmented reality. Digital content (user-generated content) is offering challenges and

opportunities for tourism destinations, mainly the uses of and the influence on consumer behavior. Academic research should follow the developments in these fields and keep up the pace to explore issues and suggest strategies.

Previous research [6] pointed out that studies on strategic principles, underpinned by the understanding of cause–effect relationships, are rare. It is believed that exciting opportunities for future research include increased attention on enabling promises made to tourists and development of strategic and research principles and an increased focus on the study of actual behavior [6]. The paper by Ávila-Robinson and Wakabayashi [9] also shows that additional value will come from research that integrates up-to-date destination management and marketing topics in a blended and interdisciplinary approach.

Future research projects could include the investigation of collaboration in destination marketing, the role of stakeholders and governance in destination marketing, and its return on investment. Another interesting pathway for research endeavors is to explore the future based on a scenario approach of destination management and marketing within a context of globalization, digitalization, and uncertainty due to various natural and health crises. Studies could attempt to identify and analyze the key issues, challenges, and trends that are impacting upon destinations; explore and identify the characteristics and future role and functions in the field of marketing such as tourism network hubs and digital content masters and facilitators; and suggest suitable strategies to address managerial and marketing issues.

Further research is also needed for better understanding of the contextual aspects determining the success and effectiveness of sustainable and environmentally friendly interventions [254]. Lastly, there is also a need for more conceptual and empirical studies in performance measurement and evaluation.

**Funding:** This research received no external funding.

**Conflicts of Interest:** The author declares no conflict of interest.

**Entry Link on the Encyclopedia Platform:** https://encyclopedia.pub/4754.

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
