# Peer review of "Tourism Destination Marketing: Academic Knowledge"

_encyclopedia, doi:10.3390/encyclopedia1010007_

Round 1

Reviewer 1 Report

I have review the manuscript presented as a review of tourism destination marketing. 

First, I would recommend to change the title to be more "attractive" - just as it is, the reader does not know what to expect until starts to read it.

Then, there are a few unclear issues:

1) I have expected to find a systematic review - as mentioned in line 45 - what is presented does not follow the steps of this type of review. Thus, my question is: what methodology/procedure was used?

2) I have also expected to find a larger reference list considering that a period of three decades is mentioned  as purpose - so, please consider to extend it. The literature in the field is rather broad.

3) Table 1 reports topics and issues proposed by the authors or taken from reference [1]? Authors could add value by adding references with examples to each topic from table 1. 

4) The introduction part could be extended by emphasizing the need of this review in the field. 

Author Response

Many thanks for your comments and suggestions.

Responses are on the attached file.

Reviewer 2 Report

The entry keeps a perfect balance between detailed information and sythesis. Also the content is well balanced. Marketing is always connected with two main issues: the content and the tools used. In the case of TDM another important issue arises: organizational context. I found the two earlier presented in completely satisfactory level. I found the content of the table especially educating. However, I can see some options for improvements regarding the organizational aspect. Nowadays the TDM is mainly a collaborative marketing and this is underlined in the text. Still, in my opinion, there is a room for extending this issue. It is true that operations of DMOs are practical implementations of collaborative marketing principles, but collaboration between stakeolders in their marketing activities is not to be limited to DMOs. The wider presentation of the importance of networking in TDM is highly welcomed. Probably, some publications by A. Fyall and B. Garrod can be useful.

Author Response

(The authors gave the same response as above.)

Round 2

Reviewer 1 Report

The manuscript was improved according to the recommendations. 

Reviewer 2 Report

Really well done. I'm impressed.